# HW-NAS-Bench: HardWare-aware Neural Architecture Search Benchmark

**Chaojian Li, Zhongzhi Yu, Yonggan Fu, Yongan Zhang, Yang Zhao, Haoran You, Qixuan Yu, Yue Wang & Yingyan Lin**
Department of Electrical and Computer Engineering
Rice University
{cl114,zy42,yf22,yz87,zy34,hy34,qy12,yw68,yingyan.lin}@rice.edu

## Abstract

HardWare-aware Neural Architecture Search (HW-NAS) has recently gained tremendous attention by automating the design of deep neural networks deployed in more resource-constrained daily life devices. Despite its promising performance, developing optimal HW-NAS solutions can be prohibitively challenging as it requires cross-disciplinary knowledge in the algorithm, micro-architecture, and device-specific compilation. First, to determine the hardware-cost to be incorporated into the NAS process, existing works mostly adopt either pre-collected hardware-cost look-up tables or device-specific hardware-cost models. The former can be time-consuming due to the required knowledge of the device's compilation method and how to set up the measurement pipeline, while building the latter is often a barrier for non-hardware experts like NAS researchers. Both of them limit the development of HW-NAS innovations and impose a barrier-to-entry to non-hardware experts. Second, similar to generic NAS, it can be notoriously difficult to benchmark HW-NAS algorithms due to their significant required computational resources and the differences in adopted search spaces, hyperparameters, and hardware devices. To this end, we develop HW-NAS-Bench, the first public dataset for HW-NAS research which aims to democratize HW-NAS research to non-hardware experts and make HW-NAS research more reproducible and accessible. To design HW-NAS-Bench, we carefully collected the measured/estimated hardware performance (e.g., energy cost and latency) of all the networks in the search spaces of both NAS-Bench-201 and FBNet, on six hardware devices that fall into three categories (i.e., commercial edge devices, FPGA, and ASIC). Furthermore, we provide a comprehensive analysis of the collected measurements in HW-NAS-Bench to provide insights for HW-NAS research. Finally, we demonstrate exemplary user cases to (1) show that HW-NAS-Bench allows non-hardware experts to perform HW-NAS by simply querying our pre-measured dataset and (2) verify that dedicated device-specific HW-NAS can indeed lead to optimal accuracy-cost trade-offs. The codes and all collected data are available at https://github.com/RICE-EIC/HW-NAS-Bench.

## 1 Introduction

The recent performance breakthroughs of deep neural networks (DNNs) have attracted an explosion of research in designing efficient DNNs, aiming to bring powerful yet power-hungry DNNs into more resource-constrained daily life devices for enabling various DNN-powered intelligent functions (Ross, 2020; Liu et al., 2018b; Shen et al., 2020; You et al., 2020a). Among them, HardWare-aware Neural Architecture Search (HW-NAS) has emerged as one of the most promising techniques as it can automate the process of designing optimal DNN structures for the target applications, each of which often adopts a different hardware device and requires a different hardware-cost metric (e.g., prioritizes latency or energy). For example, HW-NAS in (Wu et al., 2019) develops a differentiable neural architecture search (DNAS) framework and discovers state-of-the-art (SOTA) DNNs balancing both accuracy and hardware efficiency, by incorporating a loss consisting of both the cross-entropy loss that leads to better accuracy and the latency loss that penalizes the network's latency on a target device.

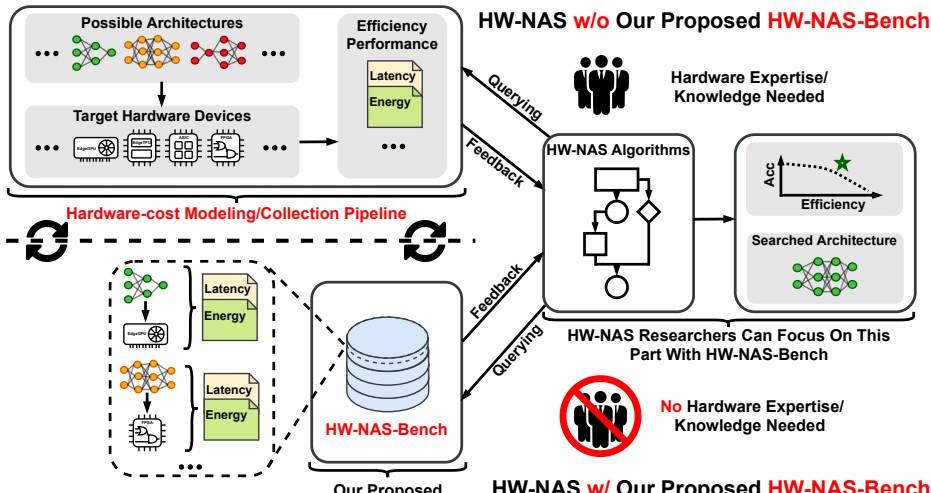

Figure 1: An illustration of our proposed **HW-NAS-Bench**

Despite the promising performance achieved by SOTA HW-NAS, there exist paramount challenges that limit the development of HW-NAS innovations. First, HW-NAS requires the collection of hardware efficiency data corresponding to (all) the networks in the search space. To do so, current practice either pre-collects these data to construct a hardware-cost look-up table or adopts device-specific hardware-cost estimators/models, both of which can be time-consuming to obtain and impose a barrier-to-entry to non-hardware experts. This is because it requires knowledge about device-specific compilation and properly setting up the hardware measurement pipeline to collect hardware-cost data. Second, similar to generic NAS, it can be notoriously difficult to benchmark HW-NAS algorithms due to the required significant computational resources and the differences in their (1) hardware devices, which are specific for HW-NAS, (2) adopted search spaces, and (3) hyperparameters. Such a difficulty is even higher for HW-NAS considering the numerous choices of hardware devices, each of which can favor very different network structures even under the same target hardware efficiency, as discussed in (Chu et al., 2020). While the number of floating-point operations (FLOPs) has been commonly used to estimate the hardware-cost, many works have pointed out that DNNs with fewer FLOPs are not necessarily faster or more efficient (Wu et al., 2019; 2018; Wang et al., 2019b). For example, NasNet-A (Zoph et al., 2018) has a comparable complexity in terms of FLOPs as MobileNetV1 (Howard et al., 2017), yet can have a larger latency than the latter due to NasNet-A (Zoph et al., 2018)'s adopted hardware-unfriendly structure.

It is thus imperative to address the aforementioned challenges in order to make HW-NAS more accessible and reproducible to unfold HW-NAS's full potential. Note that although pioneering NAS benchmark datasets (Ying et al., 2019; Dong & Yang, 2020; Klyuchnikov et al., 2020; Siems et al., 2020; Dong et al., 2020) have made a significant step towards providing a unified benchmark dataset for generic NAS works, all of them either merely provide the latency on server-level GPUs (e.g., GTX 1080Ti) or do not provide any hardware-cost data on real hardware, limiting their applicability to HW-NAS (Wu et al., 2019; Wan et al., 2020; Cai et al., 2018) which primarily targets commercial edge devices, FPGA, and ASIC. To this end, as shown in Figure 1, we develop **HW-NAS-Bench** and make the following contributions in this paper:

- We have developed **HW-NAS-Bench**, the first public dataset for HW-NAS research aiming to (1) democratize HW-NAS research to non-hardware experts and (2) facilitate a unified benchmark for HW-NAS to make HW-NAS research more reproducible and accessible, covering two SOTA NAS search spaces including NAS-Bench-201 and FBNet, with the former being one of the most popular NAS search spaces and the latter having been shown to be one of the most hardware friendly NAS search spaces.

- We provide hardware-cost data collection pipelines for six commonly used hardware devices that fall into three categories (i.e., commercial edge devices, FPGA, and ASIC), in addition to the measured/estimated hardware-cost (e.g., energy cost and latency) on these devices for all the networks in the search spaces of both NAS-Bench-201 and FBNet.

- We conduct comprehensive analysis of the collected data in HW-NAS-Bench, such as studying the correlation between the collected hardware-cost and accuracy-cost data of all

the networks on the six hardware devices, which provides insights to not only HW-NAS researchers but also DNN accelerator designers. Other researchers can extract useful insights from HW-NAS-Bench that have not been discussed in this work.

- We demonstrate exemplary user cases to show: (1) how HW-NAS-Bench can be easily used by non-hardware experts to develop HW-NAS solutions by simply querying the collected data in our HW-NAS-Bench and (2) dedicated device-specific HW-NAS can indeed lead to optimal accuracy-cost trade-offs, demonstrating the great necessity of HW-NAS benchmarks like our proposed HW-NAS-Bench.

## 2 RELATED WORKS

### 2.1 HARDWARE-AWARE NEURAL ARCHITECTURE SEARCH

Driven by the growing demand for efficient DNN solutions, HW-NAS has been proposed to automate the search for efficient DNN structures under the target efficiency constraints (Fu et al., 2020b;a; Zhang et al., 2020). For example, (Tan et al., 2019; Howard et al., 2019; Tan & Le, 2019) adopt reinforcement learning based NAS with a multi-objective reward consisting of both the task performance and efficiency, achieving promising results yet suffering from prohibitive search time/cost. In parallel, (Wu et al., 2019; Wan et al., 2020; Cai et al., 2018; Stamoulis et al., 2019) explore the design space in a differentiable manner following (Liu et al., 2018a) and significantly improve the search efficiency. The promising performance of HW-NAS has motivated a tremendous interest in applying it to more diverse applications (Fu et al., 2020a; Wang et al., 2020a; Marchisio et al., 2020) paired with target hardware devices, e.g., Edge TPU (Xiong et al., 2020) and NPU (Lee et al., 2020), in addition to the widely explored mobile phones.

As discussed in (Chu et al., 2020), different hardware devices can favor very different network structures under the same hardware-cost metric, and the optimal network structure can differ significantly when considering different application-driven hardware-cost metrics on the same hardware device. As such, it would ideally lead to the optimal accuracy-cost trade-offs if the HW-NAS design is dedicated for the target device and hardware-cost metrics. However, this requires a good understanding of both device-specific compilation and hardware-cost characterization, imposing a barrier-to-entry to non-hardware experts, such as many NAS researchers, and thus limits the development of optimal HW-NAS results for numerous applications, each of which often prioritizes a different application-driven hardware-cost metric and adopts a different type of hardware devices. As such, our proposed HW-NAS-Bench will make HW-NAS more friendly to NAS researchers, who are often non-hardware experts, as it consists of comprehensive hardware-cost data in a wide range of hardware devices for all the networks in two commonly used SOTA NAS search spaces, expediting the development of HW-NAS innovations.

### 2.2 NEURAL ARCHITECTURE SEARCH BENCHMARKS

The importance and difficulty of NAS reproducibility and benchmarking has recently gained increasing attention. Pioneering efforts include (Ying et al., 2019; Dong & Yang, 2020; Klyuchnikov et al., 2020; Siems et al., 2020; Dong et al., 2020). Specifically, NAS-Bench-101 (Ying et al., 2019) presents the first large-scale and open-source architecture dataset for NAS, in which the ground truth test accuracy of all the architectures (i.e., 423k) in its search space on CIFAR-10 (Krizhevsky et al., 2009) are provided. Later, NAS-Bench-201 (Dong & Yang, 2020) further extends NAS-Bench-101 to support more NAS algorithm categories (e.g., differentiable algorithms) and more datasets (e.g., CIFAR-100 (Krizhevsky et al., 2009) and ImageNet16-120 (Chrabaszcz et al., 2017)). Most recently, NAS-Bench-301 (Siems et al., 2020) and NATS-Bench (Dong et al., 2020) are developed to support benchmarking NAS algorithms on larger search spaces. However, all of these works either merely provide latency on the server-level GPU (e.g., GTX 1080Ti) or do not consider any hardware-cost data on real hardware at all, limiting their applicability to HW-NAS (Wu et al., 2019; Wan et al., 2020; Cai et al., 2018) that primarily targets commercial edge devices, FPGA (Wang et al., 2020b), and ASIC (Chen et al., 2016; Lin et al., 2017; 2016; Zhao et al., 2020a). This has motivated us to develop the proposed HW-NAS-Bench, which aims to make HW-NAS more accessible especially for non-hardware experts and reproducible.

A concurrent work (published after our submission) is BRP-NAS (Chau et al., 2020), which presents a benchmark for the latency of all the networks in NAS-Bench-201 (Dong & Yang, 2020) search space. In comparison, our proposed HW-NAS-Bench includes (1) more device categories (i.e., not only commercial devices, but also FPGA (Wang et al., 2020b) and ASIC (Chen et al., 2016)), (2) more hardware-cost metrics (i.e., not only latency, but also energy), and (3) more search spaces (i.e., not only NAS-Bench-201 (Dong & Yang, 2020) but also FBNet (Wu et al., 2019)). Additionally, we (4) add a detailed description of the pipeline to collect the hardware-cost of various devices and (5) analyze the necessity of device-specific HW-NAS solutions based on our collected data.

## 3 THE PROPOSED HW-NAS-BENCH FRAMEWORK

### 3.1 HW-NAS-BENCH'S CONSIDERED SEARCH SPACES

To ensure a wide applicability, our HW-NAS-Bench considers two representative NAS search spaces: (1) NAS-Bench-201's cell-based search space and (2) FBNet search space. Both contribute valuable aspects to ensure our goal of constructing a comprehensive HW-NAS benchmark. Specifically, the former enables HW-NAS-Bench to naturally integrate the ground truth accuracy data of all NAS-Bench-201's considered network architectures, while the latter ensures that HW-NAS-Bench includes the most commonly recognized hardware friendly search space.

**NAS-Bench-201 Search Space.** Inspired from the search space used in the most popular cell-based NAS, NAS-Bench-201 adopts a fixed cell search space, where each architecture consists of a predefined skeleton with a stack of the searched cell that is represented as a densely-connected directed acyclic graph (DAG). Specifically, it considers 4 nodes and 5 representative operation candidates for the operation set, and varies the feature map sizes and the dimensions of the final fully-connected layer to handle its considered three datasets (i.e., CIFAR-10, CIFAR-100 (Krizhevsky et al., 2009), and ImageNet16-120 (Chrabaszcz et al., 2017)), leading to a total of $3 \times 5^6 = 46875$ architectures. Training log and accuracy are provided for each architecture. However, NAS-Bench-201 can not be directly used for HW-NAS as it only includes theoretical cost metrics (i.e., FLOPs and the number of parameters (#Params)) and the latency on a server-level GPU (i.e., GTX 1080Ti). **HW-NAS-Bench enhances NAS-Bench-201 by providing all the $46875$ architectures' measured/estimated hardware-cost on six devices, which are primarily targeted by SOTA HW-NAS works**.

**FBNet Search Space.** FBNet (Wu et al., 2019) constructs a layer-wise search space with a fixed macro-architecture, which defines the number of layers and the input/output dimensions of each layer and fixes the first and last three layers with the remaining layers to be searched. In this way, the network architectures in the FBNet (Wu et al., 2019) search space have more regular structures than those in NAS-Bench-201, and have been shown to be more hardware friendly (Fu et al., 2020a; Ma et al., 2018). The 9 considered pre-defined cell candidates and 22 unique positions lead to a total of $9^{22} \approx 10^{21}$ unique architectures. While HW-NAS researchers can develop their search algorithms on top of the FBNet (Wu et al., 2019) search space, tedious efforts are required to build the hardware-cost look-up tables or models for each target device. **HW-NAS-Bench provides the measured/estimated hardware-cost on six hardware devices for all the $10^{21}$ architectures in the FBNet search space, aiming to make HW-NAS research more friendly to non-hardware experts and easier to be benchmarked**.

### 3.2 HARDWARE-COST COLLECTION PIPELINE AND THE CONSIDERED DEVICES

To collect the hardware-cost data for all the architectures in both the NAS-Bench-201 and FBNet search spaces, we construct a generic hardware-cost collection pipeline (see Figure 2) to automate the process. The pipeline mainly consists of the target devices and corresponding deployment tools (e.g., compilers). Specifically, it takes all the networks as its inputs, and then compiles the networks to (1) convert them into the device's required execution format and (2) optimize the execution flow, the latter of which aims to optimize the hardware performance on the target devices. For example, for collecting the hardware-cost in an Edge GPU, we first set the device in the Max-N mode to fully make use of all available resources following (Wofk et al., 2019), and then set up the embedded power rail monitor (Texas Instruments Inc.) to obtain the real-measured latency and energy via sysfs (Patrick Mochel and Mike Murphy.), averaging over 50 runs. We can see that **the hardware-cost collection pipeline requires various hardware domain knowledge, includ-**

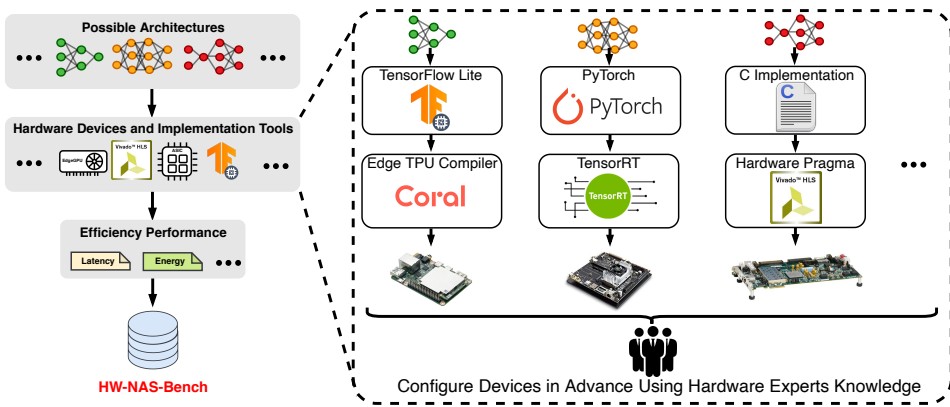

Figure 2: Illustrating the hardware-cost collection pipeline applicable to various hardware devices.

**ing machine learning development frameworks, device compilation, embedded systems, and device measurements**, imposing a barrier-to-entry to non-hardware experts.

Next, we briefly introduce the six considered hardware devices (as summarized in Table 1) and the specific configuration required to collect the hardware-cost data on each device.

**Edge GPU:** NVIDIA Edge GPU Jetson TX2 (Edge GPU) is a commercial device with a 256-core Pascal GPU and a 8GB LPDDR4, targeting IoT applications (NVIDIA Inc., a). When plugging an Edge GPU into the above hardware-cost collection pipeline, we first compile the network architectures in both NAS-Bench-201 and FBNet spaces to (1) convert them to the TensorRT format and (2) optimize the inference implementation within NVIDIA's recommended TensorRT runtime environment, and then execute them in the Edge GPU to measure the consumed energy and latency.

**Raspi 4:** Raspberry Pi 4 (Raspi 4) is the latest Raspberry Pi device (Raspberry Pi Limited.), consisting of a Broadcom BCM2711 SoC and a 4GB LPDDR4. To collect the hardware-cost operating on it, we compile the architecture candidates to (1) convert them into the TensorFlow Lite (TFLite) (Abadi et al., 2016) format and (2) optimize the implementation using the official interpreter (Google LLC., 2020) in Raspi 4, where the interpreter will be pre-configured.

**Edge TPU:** An Edge TPU Dev Board (Edge TPU) (Google LLC., a) is a dedicated ASIC accelerator developed by Google, targeting Artificial Intelligence (AI) inference for edge applications. Similar to the case when using Raspi 4, all the architectures are converted into the TFLite format. After that, an Edge TPU compiler will be used to convert the pre-built TFLite model into a more compressed format which is compatible to the pre-configured runtime environment in the Edge TPU.

**Pixel 3:** Pixel 3 is one of the latest Pixel mobile phones (Google LLC., e), which are widely used as the target platforms by recent NAS works (Xiong et al., 2020; Howard et al., 2019; Tan et al., 2019). To collect the hardware-cost in Pixel 3, we first convert all the architectures into the TFLite format, then use TFLite's official benchmark binary file to obtain the latency, when configuring the Pixel 3 device to only use its big cores for reducing the measurement variance as in (Xiong et al., 2020; Tan et al., 2019).

**ASIC-Eyeriss:** For collecting the hardware-cost data in ASIC, we consider a SOTA ASIC accelerator, Eyeriss (Chen et al., 2016). Specifically, we adopt the SOTA ASIC accelerator's performance simulators: (1) Accelergy (Wu et al., 2019)+Timeloop (Parashar et al., 2019) and (2) DNN-Chip Predictor (Zhao et al., 2020b), both of which automatically identify the optimal algorithm-to-hardware mapping methods for each architecture and then provide the estimated hardware-cost of the network execution in Eyeriss.

Table 1: Important details about the six hardware devices considered by our HW-NAS-Bench.

| Devices | Edge GPU | Raspi 4 | Edge TPU | Pixel 3 | ASIC-Eyeriss | FPGA |
|---|---|---|---|---|---|---|
| **Collected Metrics** | Latency (ms) Energy (mJ) | Latency (ms) | Latency (ms) | Latency (ms) | Latency (ms) Energy (mJ) | Latency (ms) Energy (mJ) |
| **Collecting Method** | Measured | Measured | Measured | Measured | Estimated | Estimated |
| **Runtime Environment** | TensorRT | TensorFlow Lite | Edge TPU Runtime | TensorFlow Lite | Accelergy+Timeloop / DNN-Chip Predictor | Vivado HLS |
| **Customizing Hardware?** | ✗ | ✗ | ✗ | ✗ | ✓ | ✓ |
| **Category** | | Commercial Edge Devices | | | ASIC | FPGA |

Table 2: Two types of correlation coefficients (larger means more correlated) between the real-measured hardware-cost of the whole architectures and the approximated hardware-cost based on 100 randomly sampled architectures from the FBNet search space.

| Correlation Coefficient Types | Datasets | Latency on Edge GPU | Energy on Edge GPU | Latency on Raspi 4 | Latency on Edge TPU | Latency on Pixel 3 |
|---|---|---|---|---|---|---|
| Pearson Correlation Coefficient | CIFAR-100 | 0.9200 | 0.9116 | 0.9219 | 0.4935 | 0.9324 |
| | ImageNet | 0.8634 | 0.9640 | 0.9897 | 0.7153 | 0.9162 |
| Kendall Rank Correlation Coefficient | CIFAR-100 | 0.7373 | 0.7240 | 0.7470 | 0.3551 | 0.8593 |
| | ImageNet | 0.7111 | 0.8379 | 0.9163 | 0.5806 | 0.8064 |

**FPGA:** FPGA is a widely adopted AI acceleration platform featuring a higher hardware flexibility than ASIC and more decent hardware efficiency than commercial edge devices. To collect hardware-cost data in this platform, we first develop a SOTA chunk based pipeline structure (Shen et al., 2017; Zhang et al., 2020) implementation, compile all the architectures using the standard Vivado HLS toolflow (Xilinx Inc., a), and then obtain the hardware-cost on a Xilinx ZC706 board with a Zynq XC7045 SoC (Xilinx Inc., b).

More details about the pipeline for each of the aforementioned devices are provided in the Appendix D for better understanding.

In our HW-NAS-Bench, to estimate the hardware-cost of the networks in the FBNet search space (Wu et al., 2019) when being executed on the commercial edge devices (i.e., Edge GPU, Raspi 4, Edge TPU, and Pixel 3), we sum up the hardware-cost of all unique blocks (i.e., "block" in the FBNet space (Wu et al., 2019)) within the network architectures. To validate that such an approximation is close to the corresponding real-measured results, we conduct experiments, as summarized in Table 2, to calculate two types of correlation coefficients between the measured and the approximated hardware-cost based on 100 randomly sampled architectures from the FBNet search space. We can see that our approximated hardware-cost is highly correlated with the real-measured one, except for the case on the Edge TPU, which we conjecture is caused by the adopted in-house Edge TPU compiler (Google LLC., c). More visualization results can be found in the Appendix A.

## 4 ANALYSIS ON HW-NAS-BENCH

In this section, we provide analysis and visualization of the hardware-cost and corresponding accuracy data (the latter only for architectures in NAS-Bench-201) for all the architectures in the two considered search spaces. Specifically, our analysis and visualization confirm that (1) commonly used theoretical hardware-cost metrics such as FLOPs do not correlate well with the measured/estimated hardware-cost; (2) hardware-cost of the same architectures can differ a lot when executed on different devices; and (3) device-specific HW-NAS is necessary because optimal architectures resulting from HW-NAS targeting on one device can perform poorly in terms of the hardware-cost when being executed on another device.

### 4.1 CORRELATION BETWEEN COLLECTED HARDWARE-COST AND THEORETICAL ONES

To confirm whether commonly used theoretical hardware-cost metrics align with real-measured/estimated ones, we summarize the calculated correlation between the collected hardware-cost in our HW-NAS-Bench and the theoretical metrics (i.e., FLOPs and #Params), based on the data for all the architectures in both search spaces on all the six considered hardware devices where a total of four different datasets are involved.

Table 3: Kendall Rank Correlation Coefficient between real-measured/estimated hardware-cost and theoretical ones considering the **NAS-Bench-201 search space**, where coefficients <0.5 are **bolded**.

| Dataset | Metrics | Edge GPU Latency | Edge GPU Energy | Raspi 4 Latency | Edge TPU Latency | Pixel 3 Latency | ASIC-Eyeriss Latency | ASIC-Eyeriss Energy | FPGA Latency | FPGA Energy |
|---|---|---|---|---|---|---|---|---|---|---|
| CIFAR-10 | FLOPs | **0.3571** | **0.4064** | 0.7394 | **0.1847** | 0.6823 | **0.4178** | 0.5359 | 0.8313 | 0.8313 |
| | #Params | **0.3571** | **0.4064** | 0.7394 | **0.1847** | 0.6823 | **0.4178** | 0.5359 | 0.8313 | 0.8313 |
| CIFAR-100 | FLOPs | **0.3589** | **0.4073** | 0.7384 | **0.1851** | 0.6844 | **0.4197** | 0.5360 | 0.8313 | 0.8313 |
| | #Params | **0.3589** | **0.4073** | 0.7384 | **0.1851** | 0.6844 | **0.4197** | 0.5360 | 0.8313 | 0.8313 |
| ImageNet16-120 | FLOPs | **0.3544** | **0.3868** | 0.6303 | **0.2635** | 0.7017 | **0.4166** | 0.5363 | 0.9205 | 0.9205 |
| | #Params | **0.3544** | **0.3868** | 0.6303 | **0.2635** | 0.7017 | **0.4166** | 0.5363 | 0.9205 | 0.9205 |

Table 4: Kendall Rank Correlation Coefficient between real-measured/estimated hardware-cost and theoretical ones considering the **FBNet search space**, where coefficients <0.5 are **bolded**.

| Dataset | Metrics | Edge GPU | | Raspi 4 | Pixel 3 | ASIC-Eyeriss | | FPGA | |
|---|---|---|---|---|---|---|---|---|---|
| | | Latency | Energy | Latency | Latency | Latency | Energy | Latency | Energy |
| CIFAR-100 | FLOPs | **0.0149** | **0.1564** | 0.7713 | 0.8092 | 0.8490 | 0.7854 | 0.8710 | 0.8710 |
| | #Params | **-0.0733** | **0.0202** | **0.4910** | **0.3734** | **0.4297** | 0.6455 | 0.5151 | 0.5151 |
| ImageNet | FLOPs | **0.4633** | 0.6094 | 0.7531 | 0.7678 | 0.8935 | 0.7970 | 0.8643 | 0.8643 |
| | #Params | **0.0985** | **0.1840** | **0.2318** | **0.2357** | **0.3202** | **0.4140** | **0.4198** | **0.4198** |

As summarized in Tables 3 - 4, commonly used theoretical hardware-cost metrics (i.e., FLOPs and #Params) do not always correlate well with measured/estimated hardware-cost for the architectures in both the NAS-Bench-201 and FBNet spaces. For example, there exists at least one coefficient <0.5 on all devices, especially for the cases with real-measured/estimated hardware-cost on commonly considered edge platforms including Edge GPU, Edge TPU, and ASIC-Eyeriss. As such, HW-NAS based on the theoretical hardware-cost might lead to sub-optimal results, motivating HW-NAS benchmarks like our HW-NAS-Bench. Note that we consider the Kendall Rank Correlation Coefficients (Abdi, 2007), which is a commonly used correlation coefficient in both recent NAS frameworks and benchmarks (You et al., 2020b; Siems et al., 2020; Yang et al., 2020).

## 4.2 CORRELATION AMONG COLLECTED HARDWARE-COST ON DIFFERENT DEVICES

To check how much the hardware-cost of the same architectures on different devices correlate, we visualize the correlation between the hardware-cost collected from every two paired devices based on the data for all the architectures in both the NAS-Bench-201 and FBNet search spaces with each of the architectures associated with 9 different hardware-cost metrics.

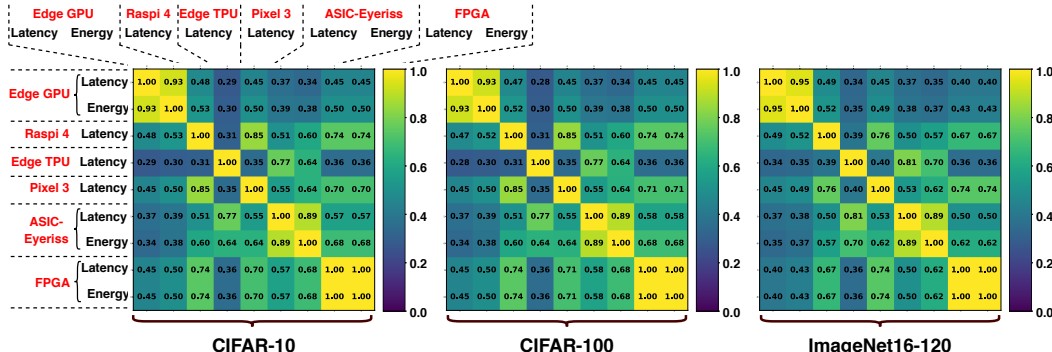

Figure 3: Kendall Rank Correlation Coefficient between real-measured/estimated hardware-cost in different devices considering the **NAS-Bench-201 search space**.

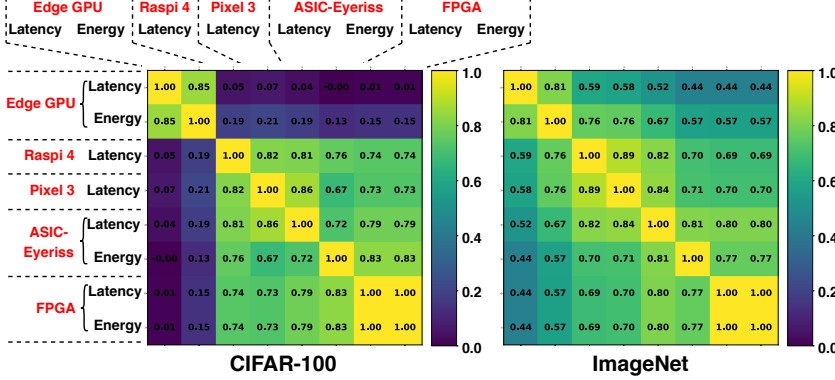

Figure 4: Kendall Rank Correlation Coefficient between real-measured/estimated hardware-cost in different devices considering the **FBNet search space**.

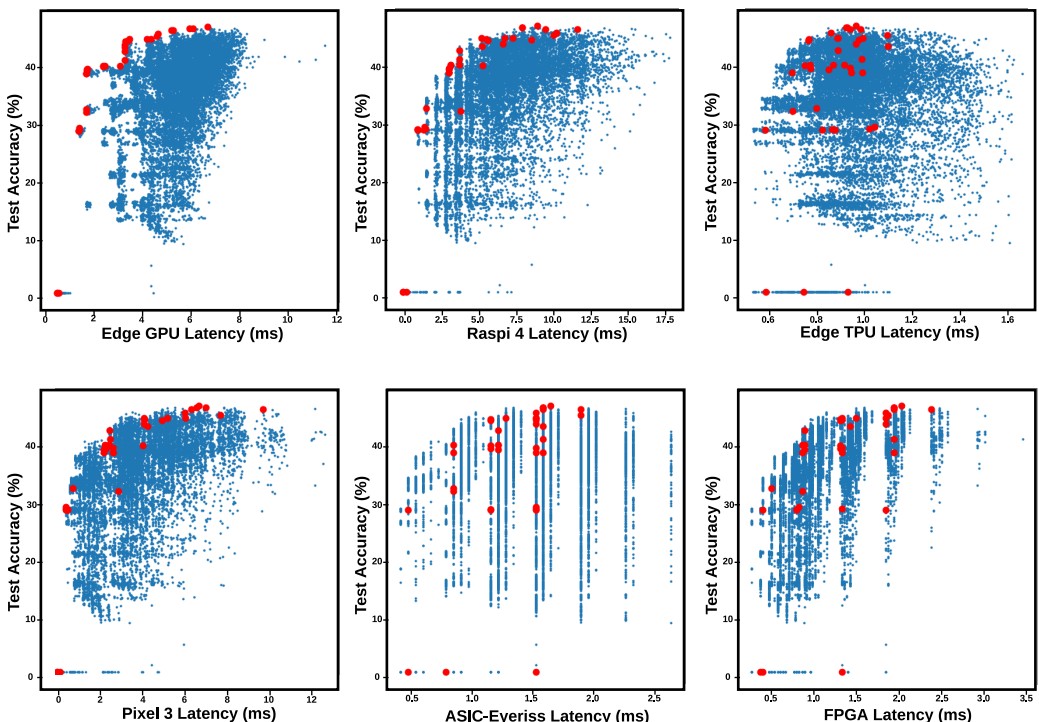

Figure 5: Accuracy vs. hardware-cost on different devices considering NAS-Bench-201, where points in red denote the architectures with the optimal trade-offs between "accuracy on ImageNet16-120 vs. latency measured on Edge GPU", of which the architectures represent the ground truth of HW-NAS targeting Edge GPUs.

The visualization in Figures 3 - 4 indicates that hardware-cost of the same network architectures can differ a lot when being executed on different devices. More specifically, the correlation coefficients can be as small as -0.00 (e.g., Edge GPU latency vs. ASIC-Eyeriss energy for the architectures in the FBNet search space), which is resulting from the large difference in their underlying (1) hardware micro-architectures and (2) available hardware resources. Thus, the resulting architecture of HW-NAS targeting one device might perform poorly when being executed on other devices, motivating device-specific HW-NAS; Furthermore, it is crucial to develop comprehensive hardware-cost datasets like our HW-NAS-Bench to enable fast development and ensure optimal results of HW-NAS for different applications.

### 4.3 OPTIMAL ARCHITECTURES ON DIFFERENT HARDWARE DEVICES

To confirm the necessity of performing device-specific HW-NAS from another perspective, we summarize the test accuracy vs. hardware-cost of all the architectures in NAS-Bench-201 considering the ImageNet16-120 dataset, and analyze the architectures with the optimal accuracy-cost trade-offs for different devices.

As shown in Figure 5, such optimal architectures for different devices are not the same. For example, the optimal architectures on Edge GPU (marked as red points) can perform poorly in terms of the hardware-cost in other devices, especially in ASIC-Eyeriss and Edge TPU whose hardware-cost exactly has the smallest correlation coefficient with the hardware-cost measured in Edge GPU, which is shown in Figure 3. Again, this set of analysis and visualization confirms that HW-NAS targeting on one device can perform poorly in terms of the hardware-cost when being executed on another device, thus motivating the necessity of device-specific HW-NAS.

## 5 USER CASES: BENCHMARK SOTA HW-NAS ALGORITHMS

In this section, we will demonstrate the user cases of our HW-NAS-Bench to show (1) how non-hardware experts can use it to develop HW-NAS solutions by simply querying the hardware-cost

Table 5: Inference accuracy and latency comparison of the optimal architectures resulting from HW-NAS-Bench when targeting different hardware devices.

| Targeted Device in HW-NAS | Top-1 Acc.(%) | Latency on Edge GPU (ms) | Latency on Raspi 4 (ms) | Latency on FPGA (ms) |
|---|---|---|---|---|
| Edge GPU | 74.11 | **9.96** | 31.01 | 20.19 |
| Raspi 4 | 73.46 | 13.88 | **22.91** | 15.39 |
| FPGA | 73.51 | 20.65 | 25.43 | **13.96** |

data and (2) dedicated device-specific HW-NAS can indeed often lead to optimal accuracy-cost trade-offs, again showing the important need for HW-NAS benchmarks like our HW-NAS-Bench to enable more optimal HW-NAS solutions via device-specific HW-NAS.

**Benchmark Setting.** We adopt a SOTA HW-NAS algorithm, ProxylessNAS (Cai et al., 2018) for this experiment. As an example to use our HW-NAS-Bench, we use ProxylessNAS to search over the FBNet (Wu et al., 2019) search space on CIFAR-100 (Krizhevsky et al., 2009), when targeting different devices in our HW-NAS-Bench by simply querying the corresponding device's measured/estimated hardware-cost, which has negligible overhead as compared to the HW-NAS algorithm itself, without the need for hardware expertise or knowledge during the whole HW-NAS.

### 5.1 OPTIMAL ARCHITECTURES RESULTING FROM DEVICE-SPECIFIC HW-NAS

Table 5 illustrates that the searched architectures achieve the lowest latency among all architectures when the target devices of HW-NAS are the same as the one used to measure the architecture's on-device inference latency. Specifically, when being executed on an Edge GPU, the searched architecture targeting Raspi 4 during HW-NAS leads to about a $50\%$ higher latency, while the searched architecture targeting FPGA during HW-NAS introduces over a $100\%$ higher latency, than the architecture specifically target on the Edge GPU during HW-NAS, under the same inference accuracy. This set of experiments shows that non-hardware experts can easily use our HW-NAS-Bench to develop optimal HW-NAS solutions, and demonstrates that device-specific HW-NAS is critical to guarantee the searched architectures' on-device performance.

## 6 CONCLUSION

We have developed HW-NAS-Bench, the first public dataset for HW-NAS research aiming to (1) democratize HW-NAS research to non-hardware experts and (2) facilitate a unified benchmark for HW-NAS to make HW-NAS research more reproducible and accessible. Our HW-NAS-Bench covers two representative NAS search spaces, and provides all network architectures' hardware-cost data on six commonly used hardware devices that fall into three categories (i.e., commercial edge devices, FPGA, and ASIC). Furthermore, we conduct comprehensive analysis of the collected data in HW-NAS-Bench, aiming to provide insights to not only HW-NAS researchers but also DNN accelerator designers. Finally, we demonstrate exemplary user cases of HW-NAS-Bench to show: (1) how HW-NAS-Bench can be easily used by non-hardware experts via simply querying the collected data to develop HW-NAS solutions and (2) dedicated device-specific HW-NAS can indeed lead to optimal accuracy-cost trade-offs, demonstrating the great necessity of HW-NAS benchmarks like our proposed HW-NAS-Bench. It is expected that our HW-NAS-Benchcan significantly expedite and facilitate HW-NAS research innovations.

ACKNOWLEDGEMENT

The work is supported by the National Science Foundation (NSF) through the CNS Division of Computer and Network Systems (Award number: 2016727).

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

## A MORE VISUALIZATION ON THE MEASURED HARDWARE-COST FOR THE FBNET SEARCH SPACE

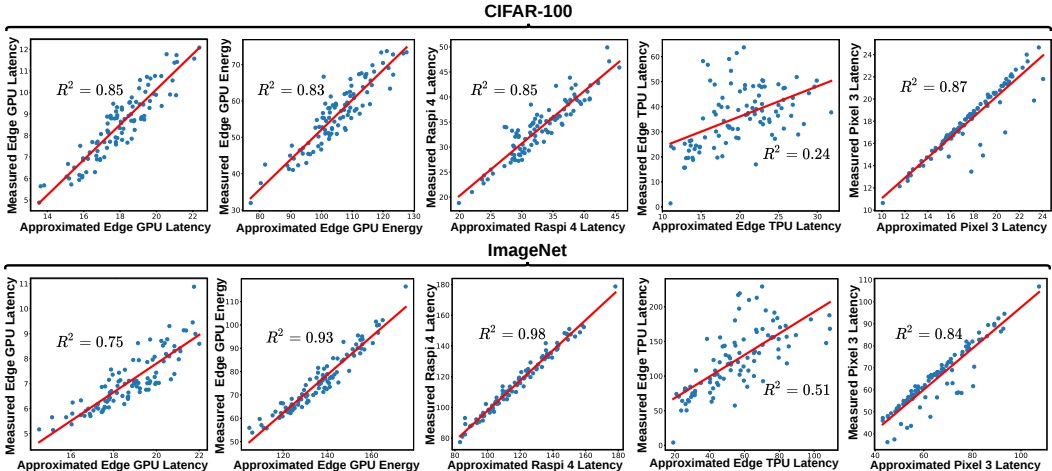

Figure 6: Comparison between the **approximated** and **measured** hardware-cost on CIFAR-100 (Top) and ImageNet (Bottom), where the red line indicates the fitting line for all the measured data, and $R^2$ represents the square of the Pearson Correlation Coefficient (Benesty et al., 2009).

Fig. 6 shows a comparison between the approximated and measured hardware-cost of randomly sampled 100 architectures when being executed on commercial edge devices using the ImageNet and CIFAR-100 datasets, which verifies that our approximation of summing up the performance of the unique blocks is a simple yet quite accurate for providing the hardware-cost for networks in the FBNet space and is consistent with our observation in Table 2.

## B COMPARING THE ESTIMATED COST EXECUTED ON EYERISS USING ACCELERGY+TIMELOOP AND DNN-CHIP REDICTOR

Both Accelergy (Wu et al., 2019)+Timeloop (Parashar et al., 2019) and DNN-Chip Predictor (Zhao et al., 2020b) are able to simulate the latency and energy cost of Eyeriss (Chen et al., 2016), a SOTA ASIC DNN accelerator, when giving the network architectures. From Table 6, they nearly give the same estimation for the latency and energy cost: specifically, the mean of their differences is 6.096%, the standard deviation of the differences is 0.779%, the Pearson correlation coefficient is 0.9998, and the Kendall Rank correlation coefficient is 0.9633, in term of the average performance, when being benchmarked with NAS-Bench-201 on 3 datasets. Therefore, we use the average value of their predictions as the estimated latency and energy on Eyeriss in our proposed **HW-NAS-Bench**.

Table 6: The differences of the hardware-cost estimation given by Accelergy (Wu et al., 2019)+Timeloop (Parashar et al., 2019) and DNN-Chip Predictor (Zhao et al., 2020b), considering NAS-Bench-201 on 3 datasets.

| Datasets | Hardware-cost | Mean of Differences | Standard Deviation of Differences | Pearson Correlation Coefficient | Kendall Rank Correlation Coefficient |
|---|---|---|---|---|---|
| CIFAR-10 | Latency | 1.648% | 0.642% | 0.9999 | 0.9888 |
|  | Energy | 10.96% | 1.035% | 0.9997 | 0.9374 |
| CIFAR-100 | Latency | 1.572% | 0.611% | 1.0000 | 0.9888 |
|  | Energy | 10.93% | 1.029% | 0.9997 | 0.9374 |
| ImageNet16-120 | Latency | 1.338% | 0.520% | 0.9999 | 0.9888 |
|  | Energy | 10.13% | 0.840% | 0.9998 | 0.9388 |
| Average Performance |  | 6.096% | 0.779% | 0.9998 | 0.9633 |

Table 7: **Left**: the marco-architectures of the search space proposed in the FBNet (Wu et al., 2019) for the ImageNet classification; **Right**: our modified search space to fit the input image size of the CIFAR-100 dataset. In the tables, "TBS" means the layer type needs to be searched and "Stride" denotes the stride of the first block in the stage. Here the modified parameters are emphasized as bold characters.

| Input Shape | Block | Filter# | Block# | Stride | Input Shape | Block | Filter# | Block# | Stride |
|---|---|---|---|---|---|---|---|---|---|
| $224^2 \times 3$ | $3 \times 3$ conv | 16 | 1 | 2 | $\mathbf{32}^2 \times 3$ | $3 \times 3$ conv | 16 | 1 | **1** |
| $112^2 \times 16$ | TBS | 16 | 1 | 1 | $\mathbf{32}^2 \times 16$ | TBS | 16 | 1 | 1 |
| $112^2 \times 16$ | TBS | 24 | 4 | 2 | $\mathbf{32}^2 \times 16$ | TBS | 24 | 4 | **1** |
| $56^2 \times 24$ | TBS | 32 | 4 | 2 | $\mathbf{32}^2 \times 24$ | TBS | 32 | 4 | 2 |
| $28^2 \times 32$ | TBS | 64 | 4 | 2 | $\mathbf{16}^2 \times 32$ | TBS | 64 | 4 | 2 |
| $14^2 \times 64$ | TBS | 112 | 4 | 1 | $\mathbf{8}^2 \times 64$ | TBS | 112 | 4 | 1 |
| $14^2 \times 112$ | TBS | 184 | 4 | 2 | $\mathbf{8}^2 \times 112$ | TBS | 184 | 4 | 2 |
| $7^2 \times 184$ | TBS | 352 | 1 | 1 | $\mathbf{4}^2 \times 184$ | TBS | 352 | 1 | 1 |
| $7^2 \times 352$ | $1 \times 1$ conv | 1984 | 1 | 1 | $\mathbf{4}^2 \times 352$ | $1 \times 1$ conv | 1504 | 1 | 1 |
| $7^2 \times 1984$ | $7 \times 7$ avgpool | - | 1 | 1 | $\mathbf{4}^2 \times 1504$ | $\mathbf{4 \times 4}$ avgpool | - | 1 | 1 |
| 1504 | fc | 1000 | 1 | - | 1504 | fc | 100 | 1 | - |

## C   MINOR MODIFICATIONS ON THE FBNET SEARCH SPACE WHEN BENCHMARKING ON CIFAR-100

Here we describe our modification on the FBNet search space when benchmarking on CIFAR-100 (i.e., the setting in Section 5) by comparing the marco-architectures before and after such modification in Table 7.

## D   DETAILS OF THE PIPELINE USED TO COLLECT HARDWARE-COST DATA

### D.1   COLLECT PERFORMANCE ON THE EDGE GPU

NVIDIA Edge GPU Jetson TX2 (Edge GPU) (NVIDIA Inc., a) is a commonly used commercial edge device, consisting of a quad-core Arm Cortex-A57, a dual-core NVIDIA Denver2, a 256-core Pascal GPU, and a 8GB 128-bit LPDDR4, for various deep learning applications including classification (Li et al., 2020), segmentation (Siam et al., 2018), and depth estimation (Wofk et al., 2019), targeting IoT, and self-driving environments. Although widely-used TensorFlow (Abadi et al., 2016) and PyTorch (Paszke et al., 2019) can be directly used in Edge GPUs, to achieve faster inference, TensorRT (NVIDIA Inc., b), a C++ library for high-performance inference on NVIDIA GPUs, is more commonly used as the runtime environment in Edge GPUs when only benchmarking inference performance (Wang et al., 2019a; NVIDIA Inc., c).

We pre-set the Edge GPU to the max-N mode to make full use of the resource on it following (Wofk et al., 2019). When plugging Edge GPUs into the hardware-cost collection pipeline, we first compile the PyTorch implementations of the network architectures in both NAS-Bench-201 and FBNet search spaces to TensorRT format models. In this way, the resulting hardware-cost can benefit from the optimized inference implementation within the TensorRF runtime environment. And then we benchmark the architectures in Edge GPUs to further measure the energy and latency using the sysfs (Patrick Mochel and Mike Murphy.) of the embedded INA3221 (Texas Instruments Inc.) power rails monitor.

### D.2   COLLECT PERFORMANCE ON RASPI 4

Raspberry Pi 4 (Raspi 4) (Raspberry Pi Limited.) is the latest Raspberry Pi device, which is a popular hardware platform for general purpose IoT applications (Zhao et al., 2015; Basu et al., 2020) and is able to support deep learning applications with specifical framework designs (Google LLC., f; Zhang et al., 2019; Geiger & Team, 2020). We choose the type of Raspi 4 with a Broadcom BCM2711 SoC and a 4GB LPDDR4 (Raspberry Pi Limited.). Similar to Edge GPUs, Raspi 4 can run architectures in the TensorFlow (Abadi et al., 2016), PyTorch (Paszke et al., 2019), or TensorFlow Lite (Google LLC., f) runtime environments. We utilize TensorFlow Lite (Google LLC., f) as it can further boost the inference efficiency.

To collect hardware-cost operating on Respi 4, an official TensorFlow Lite interpreter is pre-configured in the Raspi 4, following the settings in (Google LLC., 2020). We benchmark the possible architectures in HW-NAS-Bench on Raspi 4 after compiling them to the TensorFlow Lite (Abadi et al., 2016) format to measure the resulting latency.

### D.3    COLLECT PERFORMANCE ON THE EDGE TPU

Edge TPU (Google LLC., a) is a series of dedicated ASIC accelerators developed by Google, targeting AI inference at the edge, which can be used for classification, pose estimation, and segmentation (Xiong et al., 2020; Google LLC., b) with extremely high efficiency (e.g., 2.32× more efficient than a single SOTA desktop GPU, GTX 2080 Ti, in terms of the number of fixed-point operations per watt (Google LLC., d)). In our proposed collection pipeline, we choose the Dev Board (Google LLC., a) which provides the most functions among all products.

To collect hardware-cost in Edge TPUs, all the architectures to be benchmarked will first be converted to the TensorFlow Lite (Google LLC., f) format from their Keras (Chollet et al., 2015) implementation. After that, an in-house compiler (Google LLC., c) will be used to convert the TensorFlow Lite models into a more compressed format. This pipeline uses the least converting tools to make sure that the most operations are supported, as compared to other options (e.g., converting from the PyTorch-ONNX (Bai et al., 2020) implementation). Only the latency is collected on the Edge TPU since it lacks accurate embedded power rails monitor. We do not consider the FBNet's search space for the Edge TPU, and more details are in the Appendix A.

### D.4    COLLECT PERFORMANCE ON PIXEL 3

Pixel 3 (Google LLC., e) is one of the latest Pixel mobile phones that are widely used as the target platform by recent NAS works (Xiong et al., 2020; Howard et al., 2019; Tan et al., 2019) and machine learning framework benchmark (Google LLC., f). In our implementation, the Pixel 3 is pre-configured to use its big cores following the setting in (Xiong et al., 2020; Tan et al., 2019). Similar to the case of Raspi 4, we first convert the possible architectures in the search spaces of our proposed HW-NAS-Bench into the TensorFlow Lite format and then use the official benchmark binary files to measure the latency for each architecture.

### D.5    COLLECT PERFORMANCE ON ASIC-EYERISS

For hardware-cost data collection in ASIC, we consider Eyeriss (ASIC-Eyeriss) which is a SOTA ASIC accelerator (Chen et al., 2016). The Eyeriss chip features 168 processing elements (PEs) which are connected through a configurable dedicated on-chip network into a 2D array. A 128KB SRAM is shared by all PEs and further divided into multiple banks, each of which can be assigned to fit the input feature maps or partial sums. Thanks to these configurable hardware settings, we can adopt the optimal algorithm-to-hardware mappings for different network architectures when being executed on Eyeriss to minimize the energy or latency by maximizing data reuse opportunities for different layers.

In order to find the optimal mappings and evaluate the performance metrics on Eyeriss, we adopt SOTA performance simulators for DNN accelerators (1) Accelergy (Wu et al., 2019)+Timeloop (Parashar et al., 2019) and (2) DNN-Chip Predictor (Zhao et al., 2020b). Both of the simulators can characterize the Eyeriss's micro-architecture, perform mapping exploration, and predict the energy cost and latency metrics. Given the Eyeriss accelerator and layer information (e.g, layer type, feature map size, and kernel size) in both NAS-Bench-201 and FBNet, Accelergy+Timeloop reports the energy cost and latency characterization through an integrated mapper that finds the optimal mapping for such layer when being executed in Eyeriss. The inputs to DNN-Chip Predictor are the same as those to Accelergy+Timeloop, except that we can set the optimization metric as energy/latency/energy-delay product. DNN-Chip Predictor identifies the optimal mapping for the optimization metric and generates the estimated hardware-cost. We report the average prediction from the two simulators as the estimated hardware-cost of Eyeriss, and more details can be found in Appendix B.

Table 8: Our implemented FPGA accelerators for HW-NAS-Bench vs. SOTA FPGA accelerators, considering VGG16 on the ImageNet dataset and using Zynq XC70Z45 as the FPGA device.

| | (Zhang et al., 2018) | (Xiao et al., 2017) | **Our Implementation** |
|---|---|---|---|
| Resource Utilization | 680/900 DSP | 824/900 DSP | 723/900 DSP |
| Performance (GOP/s) | 262 | 230 | **291** |

### D.6 COLLECT PERFORMANCE ON FPGA

FPGA is a widely adopted AI acceleration platform which can offer a higher flexibility in terms of the hardware resources for accelerating AI algorithms. For collecting hardware-cost data in FPGA, we construct a SOTA chunk based pipeline structure (Zhang et al., 2018; Shen et al., 2017) as our FPGA implementation. By configuring multiple sub-accelerators (chunks) and assigning different layers to different sub-accelerators(chunks), we can balance the throughput and hardware resource consumption. To further free up our implantation's potential to reach the performance frontier across different architectures, we additionally configure hardware settings such as the number of PEs, interconnection method of PEs, and tiling/scheduling of the operations, which are commonly adopted by FPGA accelerators (Chen et al., 2017; Zhang et al., 2015; Yang et al., 2016). We then compile all the architectures using the standard Vivado HLS toolflow (Xilinx Inc., a) and obtain the bottleneck latency, the maximum latency across all sub-accelerators (chunks) of the architectures on a Xilinx ZC706 development board with Zynq XC7045 SoC (Xilinx Inc., b).

To verify our implementation, we compare our implementation's performance with SOTA FPGA accelerators (Zhang et al., 2018; Xiao et al., 2017) given the same architecture and dataset as shown in Table 8. We can see that our implementation achieves SOTA performance and thus provides insightful and trusted hardware-cost estimation for the HW-NAS-Bench.

