# OpenReview forum: "HW-NAS-Bench: Hardware-Aware Neural Architecture Search Benchmark"
_ICLR.cc/2021/Conference — ICLR 2021 Spotlight_

### Official Review · AnonReviewer2 · 2020-10-20
**Review for HW-NAS-Bench**

**Rating:** 7
**Confidence:** 5

**Review:**

### Contributions ###
* The paper proposes a benchmark for hardware-aware neural architecture search (HW-NAS). For this, the authors adopt two popular search spaces (NAS-Bench-201 and FBNet) and measure/estimate hardware performance metrics such as energy costs and latency for six hardware devices (spanning commercial edge devices, FPGA, and ASIC) for all architectures in this search spaces.
*The authors also study the rank correlation of the architectures from the  NAS-Bench-201 space regarding different hardware metrics (including both measured ones and theoretic ones like FLOPs) and find several pairs with low rank correlation. This justifies that a generic theoretic hardware metric like FLOPs is not sufficient as proxy for all practically relevant metrics.

### Significance ###
HW-NAS is an important area of research, in particular for bringing powerful DL models to edge devices and for reducing energy consumption. While the lack of generic NAS benchmarks has been addressed recently, the same did not hold true for HW-NAS. Thus, the proposed HW-NAS-Bench fills an important gap and can prove to be very useful for practitioners and HW-NAS researchers.

### Originality ###
The design of HW-NAS-Bench is mostly straight-forward in that it builds upon established search spaces and NAS benchmarks and "only" estimates hardware metrics such as latency and energy consumption on target hardware. However, this "only" of course encompasses a significant effort, in particular since six very different target hardwares are covered. I highly appreciate this effort! I hope that the author's statement "All the codes and data will be released publicly upon acceptance" also includes the code for conducting the measurements. This code would be potentially very valuable for practitioners that plan to estimate hardware costs for different search spaces or devices.

### Clarity ###
In general, summary of the design of HW-NAS-Bench and how hardware metrics are measured is outlined very clearly.
The clarification on how hardware costs for the huge FBNet search space are estimated (Appendix A) should be part of the main paper, however. Table 5 contains also relevant justification for this way of estimating. However, it is unclear to me why the authors use Pearson correlation rather than rank correlation here.

### Quality ###
The authors make a convincing case that is is not sufficient to consider theoretic hardware metrics like FLOPs for ranking different architectures since the rank correlation with respect to FLOPS and practical hardware metrics such as latency can be quite low.

However, for a NAS benchmark, the point is not so much on comparing individual architectures but rather comparing different NAS methods (that is the architectures they select from the search space). And from the paper, it is not clear that the ranking of different NAS methods would be different when using FLOPs as hardware metric compared to using  latency or energy consumption. HW-NAS-Bench is a good basis for analysing this and the paper would be strengthened by some initial results on comparing HW-NAS methods on the benchmark.

Moreover, and related to the point above, it is not really clear how to rank different NAS methods in the proposed benchmark since there is no full evaluation protocol. Two things would need clarification: (a) how would one compare Pareto fronts of different HW-NAS methods in the accuracy-hardware metric space, in particular when they intersect? (b) since there are now very many hardware metrics (latency + energy consumption for six different target devices), a way to aggregate these metrics into a single "average hardware metric" would be helpful. Without (a) and (b) it is not clear how one could actually benchmark HW-NAS methods on HW-NAS-Bench.

### Recommendation ###
In summary, I think the proposed HW-NAS-Bench will prove useful for HW-NAS development. I thus lean towards accepting the paper, in particular if the points raised above would be adressed.

### Recommendation after Author Response ###
I have read the author response and appreciate the effort spent by the authors on this response. My main criticism was addressed and the author's feedback is very convincing. The authors have not yet added this additional content to the paper. Assuming they will include it in the final version,  I am confident that this paper will meet all standards of ICLR and recommend acceptance. I increase my score accordingly to 7.

---

> ### Author Response · Authors · 2020-11-24
> **Response to Reviewer 2 (1/2)**
>
> ### Q1:Issue about “estimating hardware-cost for FBNet search space”.
> A1:
> Thank you for your great suggestions. We have moved Appendix A to the main content of the revised manuscript. The reason why we originally consider the Pearson Correlation Coefficient is to follow the metric used to benchmark hardware-cost prediction models in [1]; and following your suggestion, we have also added the Kendall Rank Correlation Coefficient to Table 2 of the revised manuscript, as summarized below.
>
> |Metrics|Latency on Edge GPU|Energy on Edge GPU|Latency on Raspi 4|Latency on Edge TPU|Latency on Pixel 3|
> |:-| :-| :-| :-| :-| :-|
> |Kendall Rank Correlation Coefficient in CIFAR-100|0.7373|0.7240|0.7470|0.3551|0.8593|
> |Kendall Rank Correlation Coefficient in ImageNet|0.7111|0.8379|0.9163|0.5806|0.8064|
>
>
> ### Q2: More results on comparing HW-NAS methods using FLOPs and hardware-cost from HW-NAS-Bench.
> A2:
> Thanks for this insightful suggestion. We agree with you that our “HW-NAS-Bench is a good basis for analyzing this (different NAS algorithms)”, and that some initial results in this regard will improve our paper. Following your suggestion, we conducted experiments to compare three recent searching algorithms adopted NAS-Bench-201: REA [2], RS [3], and REINFORCE [4], which are used to search on the NAS-Bench-201 space and CIFAR-100, using both FLOPS and a hardware metric (here: latency on an edge GPU). In particular, the accuracy and FLOPs are queried from NAS-Bench-201’s API, while the latency is queried from our HW-NAS-Bench’s API. The experiment results are summarized in the following two tables, where we make sure the accuracy achieved by the three algorithms are all around 70% for better comparing their resulting networks’ hardware performance. Interestingly, we can see that the ranking (see the last column in both tables) of the hardware latency for the three algorithms’ searched networks are different (even reverse in this case) when using FLOPs and latency, motivating the need for HW-NAS using hardware metrics provided by our HW-NAS-Bench. In our final version, we will consider more algorithms/hardware-metrics/devices to provide a more thorough comparison and discussion.
>
> Meanwhile, we would like to note that the above set of experiments are meant to provide observations when performing HW-NAS using FLOPs and hardware metrics, and the ranking of different algorithms we provided here is not sufficient to conclude those HW-NAS algorithms’ overall performance ranking because our results here are based on merely one set of experiments.  We recognize that a fair evaluation of different HW-NAS algorithms requires much more thorough benchmarking experiments based on more (1) hardware metrics, (2) edge devices, (3) NAS search spaces, (4) HW-NAS algorithms, and (5) NAS tasks, which we leave as one of our most exciting future works.
>
> |HW-NAS Algorithms|Accuracy in CIFAR-100 (%)| Latency in Edge GPU (ms)| Ranking |
> |:-:|:-:|:-:|:-:|
> |REA [2]|70.78|4.46| #2 |
> |RS [3]|70.05|4.97|  #3 |
> |REINFORCE [4]|70.50|3.36|  #1 |
>
> |HW-NAS Algorithms|Accuracy in CIFAR-100 (%)| FLOPs (M)| Ranking |
> |:-:|:-:|:-:|:-:|
> |REA [2]|70.16|51.04| #1 |
> |RS [3]|69.93|51.04|  #2 |
> |REINFORCE 4]|70.32|78.57|  #3 |
>
> ### Q3: How would one compare Pareto fronts of different HW-NAS methods in the accuracy-hardware metric space?
> A3:
> As we mentioned in our A1 to Reviewer 3, one choice is to provide a unified metric as the distance between the accuracy of the searched architecture and the architecture with the highest accuracy under the same hardware-cost (i.e., M(A, T) = A - a(T), where A and T are the searched network accuracy and hardware-cost, function a is the Pareto frontier curve achieved by HW-NAS-Bench, and M is the unified metric to benchmark between two HW-NAS algorithms), referring to the popular efficient deep learning challenge, [Low-Power Computer Vision (LPCV) Challenge](https://docs.google.com/document/d/1Rxm_N7dGRyPXjyPIdRwdhZNRye52L56FozDnfYuCi0k/edit#heading=h.9xdgszi49xgk). In this case, an average of the unified metric, computed using the data points from the Pareto fronts of different HW-NAS methods, can be used as the comparison metric.
>
> ### Q4: Aggregate metrics into a single "average hardware metric" would be helpful.
> A4: Thanks for your suggestion. We have added an “average hardware metric” which multiplies the latency and energy (if available) of all devices to our revised code release.

---

> > ### Author Response · Authors · 2020-11-24
> > **Response to Reviewer 2 (2/2)**
> >
> > *[1] Xiong, Yunyang, et al. "MobileDets: Searching for Object Detection Architectures for Mobile Accelerators." arXiv preprint arXiv:2004.14525 (2020).*
> >
> > *[2] Real, Esteban, et al. "Regularized evolution for image classifier architecture search." Proceedings of the aaai conference on artificial intelligence. Vol. 33. 2019.*
> >
> > *[3] Bergstra, James, and Yoshua Bengio. "Random search for hyper-parameter optimization." The Journal of Machine Learning Research 13.1 (2012): 281-305.*
> >
> > *[4] Williams, Ronald J. "Simple statistical gradient-following algorithms for connectionist reinforcement learning." Machine learning 8.3-4 (1992): 229-256.*

---

### Official Review · AnonReviewer3 · 2020-10-21
**Paper Review**

**Rating:** 6
**Confidence:** 3

**Review:**

**Note**: I updated my review score after the original review was written. See comments below for details.

## Summary
One important application of Neural Architecture Search is to find neural network architectures with good accuracy/inference time or accuracy/energy tradeoffs on a specific hardware device. However, the submission convincingly argues that many existing NAS benchmarks focus only on accuracy, or only provide very limited data about inference times.

In my view, the submission's main contribution is a promise to publicly release inference time and power usage measurements and code for 5-6 different hardware devices on two existing NAS benchmark tasks: NASBench-201 and FBNet. The submission also provides some analyses on this data. For example: the authors measure the correlation between inference times for the same network architectures on different hardware devices.

The authors convincingly argue that properly performing on-device inference time/energy benchmarks properly is challenging for practitioners because it "requires various hardware domain knowledge including machine learning development frameworks, device compilation, embedded systems, and device measurements." This is a major motivation for the paper and associated datasets, which present the use of pre-computed latency measurements as an easier alternative for ML researchers.

**Pros**
* **The proposed dataset seems useful for research on hardware-aware NAS algorithms.** The authors' datasets, which contain inference time measurements for several different hardware devices, should make it easier for researchers to experiment with NAS algorithms for finding better accuracy/inference time tradeoffs.
* **The paper contains some interesting analyses.** I particularly liked Figure 5; the authors identify Pareto-optimal architectures on Edge GPU and show the accuracy/inference time tradeoffs for these same devices on other hardware platforms.

**Cons**
* **The submission calls itself a "unified benchmark for HW-NAS," which may be a bit misleading.** Other NAS benchmark papers like NASBench-101 and NASBench-201 try to ensure that benchmark results from different researchers are comparable to each other; the submission does not. For example: two papers could both use the data from HW-NAS-Bench but produce incomparable results if searched for network architectures with different inference times.
* **I'm not sure what FBNet-related code is publicly available / will be released, and would like a clarification.**  While I was able to find official reproductions of specific FBNet models on github, I'm not sure whether there's an official open-source implementation of the search space itself. I'm hoping the authors can clarify, since releasing raw benchmark numbers for FBNet may not be very useful unless they're accompanied by code that can train/evaluate any architecture in the FBNet search space. If there's an official implementation, I hope the authors can provide a pointer. If the authors are using their own reproduction of the search space, I'd like to understand what they've done to verify the correctness of their implementation. (Ditto for NASBench-201 if the authors are not using the official implementation.)
* **For the FBNet search space, the information about correlations between predicted and true latency measurements is quite limited.** The author provide Pearson correlations on a random sample of architectures in Appendix A, but additional information (e.g., plotting predicted vs. true latency for a random sample of architectures) would strengthen the results. In addition, Appendix A only includes one "Pearson correlation" measurement per hardware device, and it's not clear to me whether this number is for the authors' CIFAR-10 benchmark, their ImageNet benchmark, or the union of the two. Breaking down the measurements and providing separate CIFAR-10 and ImageNet numbers would make this analysis stronger.

In addition: the usefulness of the authors' code/dataset in practice will largely depend on how easy-to-use/well-designed the code library for querying inference times is, and the paper doesn't contain enough information for me to evaluate this. This is a limitation of the review process.

**Notes on Rating:** I've given the paper a borderline score (5) in my initial review, due to the open questions mentioned in the "cons" section above. However, I believe proposed dataset could be a valuable contributions to the ML research community, and would lean toward accepting the paper if the concerns are suitably addressed.

## Experiments presented in paper
The paper promises to release on-device inference time measurements for NASBench-201, as well as a lookup-table based inference time prediction model for FBNet. For NASBench-201, measurements are provided on Edge GPU (NVIDIA Jetson TX2), Raspberry Pi 4, Edge TPU, Pixel 3, and ASIC-Eyeriss). For FBNet, latency it appears that the same devices are used, except that Edge TPU is omitted. (Although I could not find a direct explanation, Appendix A suggests that Edge TPU was excluded because a latency table-based model was not very predictive of on-device measurements.)

In addition to the raw benchmark numbers, the submission presents some experiments / sanity checks on these benchmarks (Section 4):
* Table 2: Rank correlations between model FLOPS/Parameter counts and on-device latency/energy measurements.
* Figure 3: Rank correlations for the inference latencies of the same model on different hardware devices.
* Figure 5: Taking network architectures which have pareto-optimal accuracy/latency tradeoffs on Edge GPU, and evaluating how close to optimal the network architectures are on Edge GPU in the NASBench-201 search space.

In addition, the authors present results from running three architecture searches using ProxylessNAS with different target hardware devices. (Section 5.1).

While Figures 2 and 3 and Section 5.1 mirror similar results from earlier papers like ProxylessNAS and FBNet, I still think they're valuable because they successfully validate earlier experimental claims on new search spaces and target hardware devices.

## Clarity
In general, the paper seems clear and well-organized. While the authors generally did a good job of proof-reading, I did notice a few minor typos. For example: the Section 2.1 title says "HareWare" instead of "Hardware"; in Section 3.2 under "Edge TPU", "runitime" should be changed to "runtime"; and in Appendix D, "TensorFLow" should be "TensorFlow".

## Additional Comments
The authors provide detailed information about their experimental setups in in Appendix D. I did my best to spot-check these descriptions, and the descriptions looked reasonable to me. However, I don't have enough experience with on-device benchmarks to independently certify that the benchmarks were performed correctly.

The submission includes a promise that "all the codes and data will be released publicly upon acceptance." I consider this to be a major contribution of the paper, and the paper would need to be reviewed again if this promise cannot be fulfilled for any reason.

---

> ### Comment · AnonReviewer3 · 2020-11-13
> **Thanks for providing the source code!**
>
> After looking through the source code provided by the authors, I plan to update my review score from 5 to 6. I think the code provided by the submission's authors does a good job of addressing some of the concerns from my original review ("I'm not sure what FBNet-related code is publicly available / will be released, and would like a clarification").
>
> It looks like the source code includes -- among other things -- (i) a reproduction of the FBNet search space, (ii) code for obtaining on-device benchmark numbers, (iii) a simple API for querying the inference times and energy usages of FBNet and NASBench-201 architectures, and (iv) implementations of the ProxylessNAS and FBNet search algorithms. It also looks like the API for estimating power/latency should be pretty easy to query as long as experimenters are already using PyTorch.

---

> > ### Author Response · Authors · 2020-11-21
> > **Respond to your latest comments**
> >
> > Thank you very much for taking the time to confirm our codes and for appreciating our contribution! We will provide a detailed response to your review comments soon.

---

> ### Author Response · Authors · 2020-11-24
> **Response to Reviewer 3**
>
> ### Q1:Issue about “unified benchmark for HW-NAS”
> A1:
> Sorry for not making it clear enough. It does help in “unified benchmark for HW-NAS” from the following aspects. First, as you mentioned, when two papers target the same hardware-cost in the same device, the proposed HW-NAS-Bench can provide a unified benchmark for them. Second, for the case that two papers target different hardware-cost, here we provide a potential user case example for the popular efficient deep learning challenge, [Low-Power Computer Vision (LPCV) Challenge](https://docs.google.com/document/d/1Rxm_N7dGRyPXjyPIdRwdhZNRye52L56FozDnfYuCi0k/edit#heading=h.9xdgszi49xgk), during which HW-NAS-Bench can provide an API (we will update this in the final version) for the organizers or participants to query a unified metric, i.e.,  the distance between the accuracy of the searched architecture and architecture with the highest accuracy under the same hardware-cost ( M(A, T) = A - a(T), where A and T are the searched network accuracy and hardware-cost, function a is the Pareto frontier curve achieved by HW-NAS-Bench, and M is the unified metric to benchmark between two HW-NAS algorithms).
>
> ### Q2: FBNet-related code release
> A2:
> From your latest comments, it looks like our released code has addressed your question. Furthermore, to verify the correctness of our implementation, we conduct experiments to reproduce searching for FBNet on ImageNet dataset, and the results are summarized as follow, which shows the searched networks under our codebase reach comparable performance with the ones reported in FBNet:
>
> | Model | FLOPs | Accuracy (%) |
> | :---| :---: | :---: |
> | FBNet-A | 249M | 73.0 |
> | Reproduced-FBNet-A | 252M | 72.9 |
> | FBNet-B | 295M | 74.1 |
> | Reproduced-FBNet-B | 308M | 74.3 |
> | FBNet-C | 375M | 74.9 |
> | Reproduced-FBNet-C | 432M | 75.3 |
> ### Q3: More information about the information of correlations between predicted and true latency in the FBNet search space.
> A3: Thanks for your detailed suggestion. First, we have added the suggested predicted vs. measured latency for 100 architectures randomly sampled from the FBNet search space in Section A of the Appendix as shown in the revised manuscript. Second, we would like to clarify that Table 5 in our submitted manuscript is based on the FBNet space and CIFAR-100 dataset. Third, we have added the results based on both the CIFAR-100 and ImageNet datasets to Table 2 of the revised manuscript.
>
> |Metrics|Latency on Edge GPU|Energy on Edge GPU|Latency on Raspi 4|Latency on Edge TPU|Latency on Pixel 3|
> |:-| :-| :-| :-| :-| :-|
> |Pearson Correlation Coefficient in CIFAR-100|0.9200|0.9116|0.9219|0.4936|0.9324|
> |Pearson Correlation Coefficient in ImageNet|0.8634|0.9640|0.9897|0.7153|0.9162|
> ### Q4: Minor typos.
> A4: Thanks a lot for pointing out those typos. We have fixed it in the revised paper and will more carefully proofread it in the final version.

---

### Official Review · AnonReviewer4 · 2020-10-28
**This paper analyzes the existing landscape of Neural Architecture Searches and summarizes a set of hardware and software parametrization with the aim of creating a standardized benchmark for NAS algorithms.**

**Rating:** 7
**Confidence:** 4

**Review:**

Strengths
1. Analysis of different NAS algorithm and search space
2. Comparison of measurement vs. estimation for different hardware systems

Weaknesses
1. The analyzed parameters are absolute
2. The set of analyzed hardware is limited
3. There are few mentions of the analyzed Deep Learning models

Major Comments
1. How would a configuration not present in the benchmark be handled? Via interpolation or returning an error?
2. In Section 3, how long does it take to run all the measurements and estimations? While this is a one-time cost, if the benchmark requires improvements this cost is paid again.
3. In Section 3, there are very few mentions of the analyzed models: it seems the focus is on vision networks, mostly using convolutional layers, but there are other network types which are rising to prominence and for which the hardware is optimized, such as transformers. It is suggested to add more details regarding the use of other network types, or at least analyze this different domain of NAS to provide a proper justification.
4. In Section 3.2, while the set of chosen hardware spans multiple devices and targets, it may be limited towards the “fixed” devices, such as mobile phones and edge PCs, as only a handful of them are analyzed. While these examples may be representative, they could not cover the whole search space and characteristics, limiting the applicability of the benchmark in real-world scenarios.
5. In Section 4, when analyzing the different hardware systems, there is the usage of absolute characteristics, such as FLOPs and latency, why are other relative characteristics, such as arithmetic intensity, not being considered? They could provide a better estimate and means of comparison, especially since the set of hardware is very wide, covering the whole intensity spectrum.

Related Work Suggestions
1. A. Marchisio, A. Massa, V. Mrazek, B. Bussolino, M. Martina, M. Shafique, “ NASCaps: A Framework for Neural Architecture Search to Optimize the Accuracy and Hardware Efficiency of Convolutional Capsule Networks”, to appear at The IEEE/ACM 2020 International Conference On Computer Aided Design (ICCAD), November 2020

Minor Comments
1. In Section I, in Figure 1, the text and the figures are difficult to read, as they should have a slightly bigger font size.
2. In Section 4.2, Figures 3 and 4, the axes are difficult to follow, especially since they are not repeated for the other graphs in the figures.

---

> ### Comment · AnonReviewer4 · 2020-11-20
> **The source code is valuable**
>
> The source code is comprehensive, easy to use, and potentially useful for the community

---

> > ### Author Response · Authors · 2020-11-21
> > **Respond to your latest comment.**
> >
> > Thank you very much for appreciating our efforts! We will provide a detailed response to your comments soon.

---

> ### Author Response · Authors · 2020-11-24
> **Response to Reviewer 4 (1/2)**
>
> ### Q1: How would a configuration not present in the benchmark be handled? Via interpolation or returning an error?
> A1:
> An error will be returned. Following the API design of NAS-Bench-201, an error will be returned when querying a configuration that doesn’t present in the proposed HW-NAS-Bench.
> ### Q2: In Section 3, how long does it take to run all the measurements and estimations? While this is a one-time cost, if the benchmark requires improvements this cost is paid again.
> A2:
> About one month is needed to obtain all the measurements and estimations, considering the number of devices and simulators that we currently target. Yes, this cost is to be paid again if our HW-NAS-Bench targets to incorporate hardware-cost on more emerging devices or consider more emerging HW-NAS search spaces, which we are planning to do for maintaining a long-term usefulness of our HW-NAS-Bench.
> ### Q3: Suggestions about non-vision networks.
> A3:
> Thank you for your suggestion for us to consider non-vision networks. We are collecting data from the NAS-Bench-NLP [1], which provides a benchmark for NAS in NLP tasks, and will update it once we finish the collection.  Finally, we will keep updating our HW-NAS-Bench with more devices and NAS benchmark works are released, ensuring its long-term usefulness.
> ### Q4: The set of analyzed hardware is limited.
> A4: Thank you for your good intention of helping to improve this work.
>
> (1) Regarding your comment on “fixed devices”, we humbly disagree as we have considered commonly used commercial edge devices, ASIC, and FPGA, which to our best knowledge covers most of the hardware platforms considered by SOTA HW-NAS works.  Furthermore, we would be happy to include other devices if you could kindly suggest or refer us to references with practical needs.
>
> (2) Regarding your comment of “cannot cover all and limit the applicability in real-world scenarios”,  we would like to respond from two aspects. First, to our best knowledge, the six devices that fall into three categories are representative and commonly considered hardware platforms in both the general scenarios of SOTA efficient deep learning at the edge [2 - 6] and HW-NAS [7 - 12]. Second, Reviewers 1-3 all comment that our HW-NAS-Bench covers many representative devices and can be useful in both industry and academia, e.g., R1→ “the proposed dataset/benchmark covers a significant and representative part of the most common targets for NAS algorithms”; R2→  “six very different target hardware are covered”; and R3→ “the proposed dataset seems useful for research on hardware-aware NAS algorithms”.
> ### Q5: Why are other relative characteristics, such as arithmetic intensity, not being considered?
> A5: Thanks for providing this suggestion. The reason why the arithmetic intensity is not considered is that the memory access for computing the arithmetic intensity is not straightforward to be obtained from the commercial edge devices, e.g., only peak memory footprint is accessible in Pixel 3 using TensorFlow Lite’s official benchmark tool [13]. Furthermore, the metrics we consider, i.e.,  FLOPs, #Params, latency, and energy, are more commonly used in recent NAS works [7 - 12]. Finally, following your suggestion, we computed the arithmetic intensity for ASIC-Eyeriss, and have added it to our released codes.
> ### Q6: Related Work Suggestions.
> A6: Thanks for the suggestion. We have included them in our final version.
> ### Q7: Minor Comments.
> A7: Thanks for the detailed comments. We have fixed them and will more carefully proofread the manuscript in the final version.

---

> > ### Author Response · Authors · 2020-11-24
> > **Response to Reviewer 4 (2/2)**
> >
> > *[1] Klyuchnikov, Nikita, et al. "Nas-bench-nlp: Neural architecture search benchmark for natural language processing." arXiv preprint arXiv:2006.07116 (2020).*
> >
> > *[2] Wofk, Diana, et al. "Fastdepth: Fast monocular depth estimation on embedded systems." 2019 International Conference on Robotics and Automation (ICRA). IEEE, 2019.*
> >
> > *[3] Li, Chaojian, et al. "HALO: Hardware-aware learning to optimize." European Conference on Computer Vision. Springer, Cham, 2020.*
> >
> > *[4] Siam, Mennatullah, et al. "A comparative study of real-time semantic segmentation for autonomous driving." Proceedings of the IEEE conference on computer vision and pattern recognition workshops. 2018.*
> >
> > *[5] Zhang, Jianhao, et al. "dabnn: A super fast inference framework for binary neural networks on arm devices." Proceedings of the 27th ACM International Conference on Multimedia. 2019.*
> >
> > *[6] Geiger, Lukas, and Plumerai Team. "Larq: An Open-Source Library for Training Binarized Neural Networks." Journal of Open Source Software 5.45 (2020): 1746.*
> >
> > *[7] Xiong, Yunyang, et al. "MobileDets: Searching for Object Detection Architectures for Mobile Accelerators." arXiv preprint arXiv:2004.14525 (2020).*
> >
> > *[8] Howard, Andrew, et al. "Searching for mobilenetv3." Proceedings of the IEEE International Conference on Computer Vision. 2019.*
> >
> > *[9] Tan, Mingxing, et al. "Mnasnet: Platform-aware neural architecture search for mobile." Proceedings of the IEEE Conference on Computer Vision and Pattern Recognition. 2019.*
> >
> > *[10] Bender, Gabriel, et al. "Can Weight Sharing Outperform Random Architecture Search? An Investigation With TuNAS." Proceedings of the IEEE/CVF Conference on Computer Vision and Pattern Recognition. 2020.*
> >
> > *[11] Yang, Lei, et al. "Co-Exploration of Neural Architectures and Heterogeneous ASIC Accelerator Designs Targeting Multiple Tasks." Proceedings of the 57th Annual Design Automation Conference 2020. 2020.*
> >
> > *[12] Jiang, Weiwen, et al. "Accuracy vs. efficiency: Achieving both through fpga-implementation aware neural architecture search." Proceedings of the 56th Annual Design Automation Conference 2019. 2019.*
> >
> > *[13] Google LLC. TFLite Model Benchmark Tool with C++ Binary, https://github.com/tensorflow/tensorflow/tree/master/tensorflow/lite/tools/benchmark, accessed 2020-11-14*

---

### Official Review · AnonReviewer1 · 2020-10-28
**The paper presents a dataset, HW-NAS-Bench, for evaluating neural architecture search algorithms. Based on real measurements, HW-NAS-Bench provides a valuable tool for NN designers.**

**Rating:** 7
**Confidence:** 3

**Review:**

The paper presents a benchmark / dataset, HW-NAS-Bench, for evaluating various neural architecture search algorithms. The benchmark is based on extensive measurements on real hardware. An important goal with the proposal is to support neural architecture searches for non-hardware experts. Further, the paper provides a good overview of related work in the domain.

The paper has a very good intention, i.e., to help and support non-hardware experts in the neural architecture search process. I think the paper contributes a lot to that ambition, by providing a benchmark / dataset of hardware-aware measurements / predictions that can be queried either by a person or a NAS algorithm.

The network architectures that provide the search space are NAS-Bench-201 and FBNet, and the measurements/predictions are obtained from three categories of devices, i.e., commercial edge devices, FPGAs, and ASICs). Thus, my belief is that the proposed dataset / benchmark covers a significant and representative part of the most common targets for NAS algorithms. Further, I like that the authors will publish their measurement results / estimated hardware costs for over 46000 network/hardware combinations.

The work presented in the paper is important and can potentially have a significant impact, both in industry as well as in academia.

Some other comments / questions:
* I really don't understand the idea behind Table 4. The performance is best when we run on the same hardware as we have optimized for? Am I missing something here?

---

> ### Comment · AnonReviewer1 · 2020-11-16
> **Thanks for the code and data**
>
> Thanks for providing the code and the data for your paper. I've browsed through the code and it seems well-structured and relatively easy to follow. Further, I also ran some of your examples to test the code and it worked fine. Your framework seems easy to use, and I think this piece of work can be a useful contribution to the community.

---

> > ### Author Response · Authors · 2020-11-21
> > **Respond to your latest comment.**
> >
> > Thank you very much for making the effort to confirm our codes and appreciating our contributions!

---

> ### Author Response · Authors · 2020-11-24
> **Response to Reviewer 1**
>
> ### Q1: Analysis of Table 4.
>
> A1:
> Yes, your understanding of “The performance is best when we run on the same hardware as we have optimized for” is correct, which as Reviewer 3 mentioned “mirror similar results from earlier papers like ProxylessNAS and FBNet and are valuable because they successfully validate earlier experimental claims on new search spaces and target hardware devices.” Specifically, the idea behind Table 4 includes (1) validating the necessity of device-specific HW-NAS solutions as the performance (i.e., accuracy vs. cost trade-offs) of the generated networks on a device is the best when HW-NAS is performed incorporating the hardware-cost from the same device, motivating the need for the proposed HW-NAS-Bench which contains hardware-cost of the networks from two commonly HW-NAS search spaces; and (2) an demonstration of a user case of HW-NAS-Bench where non-hardware experts can use HW-NAS-Bench to develop HW-NAS solutions by simply querying the hardware-cost, leading to competitive solutions as SOTA HW-NAS without going through the hardware-cost measurement or modeling process.

---

### Comment · ~Thomas_Chun_Pong_Chau1 · 2021-01-23
**Please cite another published dataset for hardware-aware NAS**

There is a published latency dataset on the NAS-Bench-201 search space considering a broad range of devices (desktop, mobile and embedded CPU/GPU/DSP). Please consider citing and comparing to the following work:


**BRP-NAS: Prediction-based NAS using GCNs (in NeurIPS'20)**

Paper (https://arxiv.org/abs/2007.08668)

Code and dataset (https://github.com/thomasccp/eagle)

Thank you.

---

### Decision · Program_Chairs · 2021-01-07
**Final Decision**

**Decision:**

Accept (Spotlight)

**Comment:**

This paper presents a new NAS benchmarks for hardware-aware NAS. For each of the architectures in the search space of NAS-Bench-201, it measures hardware performance (energy cost and latency) for six different hardware devices. This is extremely useful for the NAS research community, since it takes very specialized hardware domain knowledge (including machine learning development frameworks, device compilation, embedded systems, and device measurements) as well as the hardware to make these hardware-aware measurements on as many as six (very different) devices.

The code has been made available to the reviewers during the author response window and has been checked by the reviewers in the meantime. All reviewers appreciated the paper and gave (clear) acceptance scores.

Before this work, it was very hard for the average NAS researcher to assess their method properly in a hardware-aware setting, and I expect this work to change this, and to open up the very important field of hardware-aware NAS to many more researchers. For this reason I recommend to accept this paper as a spotlight.